# Impact of COVID-19 Vaccination on Short-Term Perceived Change in Physical Performance among Elite Athletes: An International Survey

**DOI:** 10.3390/vaccines11040796

**Published:** 2023-04-04

**Authors:** Olivier Bruyère, Géraldine Martens, Céline Demonceau, Axel Urhausen, Romain Seil, Suzanne Leclerc, Sébastien Le Garrec, Philippe Le Van, Pascal Edouard, Philippe M Tscholl, François Delvaux, Jean-François Toussaint, Jean-François Kaux

**Affiliations:** 1WHO Collaborating Centre for Public Health Aspects of Musculo-Skeletal Health and Ageing, Division of Public Health, Epidemiology and Health, Economics, University of Liège, 4000 Liège, Belgium; 2Department of Sport and Rehabilitation Sciences, University of Liège, 4000 Liège, Belgium; 3Physical Medicine and Sport Traumatology Department, SportS2, FIFA Medical Centre of Excellence, FIMS Collaborative Centre of Sports Medicine, University and University Hospital of Liège, 4000 Liège, Belgium; 4Réseau Francophone Olympique de la Recherche en Médecine du Sport (ReFORM) IOC Research Centre for Injury Prevention and Protection of Athlete Health, 4000 Liège, Belgium; 5Québec National Institute of Sports (INS Québec), Montréal, QC H1V 3N7, Canada; 6Luxembourg Institute of Research in Orthopedics, Sports Medicine and Science, 1460 Luxembourg, Luxembourg; 7Clinique du Sport, Centre Hospitalier de Luxembourg, 1460 Luxembourg, Luxembourg; 8Department of Orthopaedic Surgey, Centre Hospitalier de Luxembourg, 1460 Luxembourg, Luxembourg; 9National Institute of Sport, Expertise and Performance (INSEP), 75012 Paris, France; 10Sports Medicine Unit, Department of Clinical and Exercise Physiology, University Hospital of Saint-Etienne, 42055 Saint-Etienne, France; 11Inter-University Laboratory of Human Movement Biology, Université Jean Monnet, Lyon 1, Université Savoie Mont-Blanc, EA 7424, F-42023 Saint-Etienne, France; 12Department of Orthopaedic Surgery and Traumatology, Geneva University Hospitals, CH-1211 Geneva, Switzerland; 13URP 7329-IRMES (Institute for Research in Medicine and Epidemiology of Sport), National Institute of Sport, Expertise and Performance (INSEP), 75012 Paris, France

**Keywords:** SARS-CoV-2, sports, immunization, athletes, elite, athletic performance

## Abstract

COVID-19 vaccination raised concerns about its potential effects on physical performance. To assess the impact of COVID-19 vaccination on the perceived change in physical performance, we conducted an online survey among elite athletes from Belgium, Canada, France and Luxembourg, with questions about socio-demographics, COVID-19 vaccination, perceived impact on physical performance and perceived pressure to get vaccinated. Full vaccination was defined as two doses of mRNA or vector vaccine or a heterologous vaccine scheme. Among 1106 eligible athletes contacted, 306 athletes answered the survey and were included in this study. Of these, 72% perceived no change in their physical performance, 4% an improvement and 24% a negative impact following full COVID-19 vaccination. For 82% of the included athletes, the duration of the negative vaccine reactions was ≤3 days. After adjustment for potential confounding variables, practicing an individual sport, a duration of vaccine reactions longer than 3 days, a high level of vaccine reaction and the perceived pressure to get vaccinated were independently associated with a perceived negative impact on physical performance of more than 3 days after the vaccination. The perceived pressure to get vaccinated appears to be a parameter associated with the negative perceived change in the physical performance and deserves further consideration.

## 1. Introduction

The current coronavirus pandemic (COVID-19) represents an important public health challenge, requiring ambitious public health measures including vaccination. Athletes have also been impacted by COVID-19. The consecutive lockdowns prevented training in appropriate conditions and participating in sports competitions [1]. In addition, the strong incentives to get vaccinated to participate in competitions contributed to the ongoing debates and pressures around the role of vaccination in managing the pandemic [2,3]. This additional professional pressure, on top of the social one, might have impacted how athletes perceived the benefits or harms of vaccination.

The COVID-19 vaccination was associated with a broad range of local and systemic vaccine reactions reported in clinical trials and in post-marketing studies in the general population [4]. These included: local reactogenicity, headache, fatigue, fever, chills and myalgia [3,4,5]. Although generally mild, they appear to be more common in younger individuals and more pronounced after the second dose [4]. Myalgia, for instance, was reported in 21% of younger individuals following the first dose and in 37% following the second dose [3]. It could negatively impact training and performance, especially if generalized.

One of the concerns affecting compliance with vaccination in the elite athlete population was the reduction of performance due to these potential vaccine reactions [3]. The American Medical Society for Sports Medicine guidelines for vaccinating athletes mention that reactions are short-lived, particularly when compared to COVID-19 illness or subsequent quarantine/isolation requirements [6]. The risks associated with COVID-19 infection can also be greater than those related to vaccination [7,8].

However, there are very few published data on COVID-19 vaccination in high-performing athletes, notably regarding the duration of vaccine reactions and their impact on training and competition. Of the 127 British athletes preparing for the Olympic and Paralympic Games who completed a daily electronic questionnaire for 10 days following COVID-19 vaccination, 94% of the athletes reported arm pain around the injection site, lasting a median of two days [9]. Systemic reactions were reported in 70% of participants, with generalized fatigue in 28% after the first dose (median duration of 1 day) and 37% after the second dose (1 day). Still, 73% of all the athletes reported no or only a minor effect on their ability to train. Only 6% of the athletes felt unable to train; all but one returned to training after 1 day. Despite a high prevalence of systemic reactions, the deleterious impact on training was minimal. This would need to be confirmed on a larger and international sample. In addition, the aforementioned external sociological factors, such as the pressure to get vaccinated, should be considered. Indeed, in a recent study focusing on 895 Polish elite athletes’ perceptions towards vaccination, it was shown that perceived pressure from the coach and/or the sports federation was reportedly the strongest incentive to be vaccinated for 683 vaccinated athletes [10]. On the other hand, the main reason for not getting vaccinated for 212 non-vaccinated athletes was being discouraged by coaches. This underlines the important impact of pressure to get vaccinated or not by the direct professional environment. When asked about who had the greatest influence on their decision to get vaccinated or not, the second greater one, after themselves, was their relatives. This emphasizes the additional impact of social pressure on the decision to get vaccinated [10].

We, therefore, aimed to assess the impact of COVID-19 vaccination on the perceived change in physical performance among an international sample of elite athletes (i.e., meeting the 2020 Olympic Games participation standards). The secondary objectives were to assess the level and the duration of the vaccine reactions and to investigate the potential link between the pressure to get vaccinated and the physical performance perception. The influence of additional potential factors such as the type of sport (individual vs. team), the vaccine dose (first vs. second) and the level and duration of vaccine reactions was investigated, as well, in an exploratory manner.

## 2. Methods

### 2.1. Study Procedures

We conducted a retrospective cross-sectional study with data collected through an online survey completed by volunteer elite athletes from December 2021 to February 2022. The Ethics Committee of the University Teaching Hospital of Liège approved this research (CE 2021/254).

### 2.2. Participant Recruitment

The target population was recruited throughout four of the five institutions of the Réseau Francophone Olympique de la Recherche en Médecine du sport (ReFORM) IOC Research Centre in Belgium (University Hospital of Liège Sports and Traumatology Department), Canada (Institut National du Sport du Québec), France (National Institute of Sport, Exercise and Performance [INSEP]) and Luxembourg (Luxembourg Institute of Research in Orthopedics, Sports Medicine and Science [LIROMS]) [11]. The inclusion criteria were having the nationality of one of these four countries, having received a full vaccination and being on a professional sports contract at the time of enrolment, and/or meeting participation standards for the Tokyo 2020 Olympic Games. Any athletes who did not complete the entire questionnaire or those who did not receive a COVID-19 full vaccination at the time of the study were excluded. In this study, full vaccination corresponds to two doses of mRNA or vector vaccine or a heterologous vaccine scheme [12,13].

The National Olympic Committees (NOCs, Belgium, France, Luxembourg) or Institute (Québec) sent emails to their affiliated athletes with an invitation to participate in the survey, with a secured link. To guarantee anonymous participation, the investigators did not have access to the list of participants, and the NOCs/Institute only provided the total number of invited athletes. The data collected were only accessible to the investigators and not to the NOCs, affiliated institutes or athletes. Informed consent was obtained online from each participant. The survey was housed on SondageOnline (enuvo GmbH, Zurich, Switzerland) and was available in French or English.

### 2.3. Questionnaire

The full questionnaires (French and English versions) are available in Appendix A. The first version of the questionnaire was developed by a small team (OB, GM, CD and JFK) consisting of experts in sports sciences, public health, epidemiology and rehabilitation, and then reviewed by all the authors. The next version was then pre-tested by three athletes, and the updated version was considered final. The questionnaire consisted of 31 questions divided into five parts, detailed below.

COVID-19 vaccination (11 questions): This section addressed the type(s) and the time frame of the received vaccines, as well as the vaccine reactions’ intensity and duration using a 100-unit visual analog scale (VAS), where 0 corresponds to no reaction and 100 to the most intense reaction.

COVID-19 (3 questions): This section covered the testing for COVID-19, the date of a potential COVID-19-positive test and the perceived impact of COVID-19 infection on performance, using a VAS (0 = strongest deterioration in performance, 100 = highest improvement in performance, 50 meaning no effect).

Perceived physical performance (10 questions): Physical performance was assessed by a series of perceptual questions with a 100-unit VAS including: (1) the impact of vaccination on training, performance and satisfaction with performance, and; (2) the impact of being tested COVID-19-positive on performance and satisfaction with performance.

Perceived pressure to get vaccinated (3 questions): Pressure was measured using a 100-unit VAS for global pressure (societal, professional, medical) (0 = no perceived pressure, 100 = highest perceived pressure). Two other VAS were used to evaluate social and professional pressures separately.

Sociodemographic (4 questions): Sociodemographic variables were collected: age (classified as less than 18, between 18 and 25, and under 25), nationality and type of sport, and hours of sport per week (classified as between 5 and 10, 11 and 20, 21 and 35 or 35 and 50 h per week). No additional sociodemographic data were collected, as this aspect was restricted to guarantee the anonymous character.

### 2.4. Analyses

Four categorical variables were then created: (1) The sports were classified as collective or individual. (2) The durations of vaccine reactions were arbitrarily classified as no reaction, less than 1 day, between 1 and 3 days, and more than 3 days. (3) The score of the 100-unit VAS on the question “Do you feel that your full vaccination against COVID-19 has impacted your sports performances?” was used to classify the impact of full COVID-19 vaccination on physical performance as negative (0–40), without impact (41–59) or positive (60–100). (4) Being positive for COVID-19 before the first vaccine was determined by crossing the dates of positive tests and the first vaccination.

For the descriptive analyses, the quantitative variables that were normally distributed were expressed by the mean and standard deviation, and the quantitative variables that were not normally distributed were expressed by the median and percentiles (P25, P75). The qualitative variables are expressed as numbers and frequencies (%). Normal distribution was assessed using the comparison of the mean and the median, Q–Q plots, histograms and the Kolmogorov–Smirnov test.

For our primary objective (i.e., the impact of vaccination on perceived physical performance), we set an arbitrary time cut-off of 3 days of impact on performance. Based on discussion and agreement between the experts’ panel, more than 3 days of perceived consequences of vaccination would significantly impact athletes’ training. We thus created two groups of participants: group 1—those who perceived no, equal or less than 3 days’ negative impact on their performance, and group 2—those who perceived a negative impact of more than 3 days on their performance after full COVID-19 vaccination. We used Wilcoxon, chi-square or Fisher’s exact test to compare the sociodemographic variables, COVID-19 history, COVID-19 vaccination and pressure to get vaccinated between the groups.

As part of our secondary objectives, we provided descriptive analyses of the level and duration of the vaccine reactions after the first and second doses of the COVID-19 vaccination. In addition, we assessed the VAS score for global, social and professional pressure in all athletes. These pressures were also analyzed separately for the two groups of athletes who perceived no, or less than 3 days’, negative impact on their performance, and those who perceived a negative impact of more than 3 days on their performance after full COVID-19 vaccination.

In a further exploratory step, we performed a logistic regression with the perceived negative impact on physical performance during more than 3 days after each dose of vaccine (for the athletes who received two doses) as the dependent variable, adjusted with all the significant variables in the univariate analysis. A sensitivity analysis was also performed using a cut-off of 8 days (instead of 3 days) to assess the robustness of the results.

All analyses were performed with R version 3.6.1. The results were considered statistically significant when the two-tailed *p* values were <0.05.

## 3. Results

### 3.1. Participant Characteristics

A total of 1106 athletes were invited to participate; 379 responded to the survey (34.3%) and 306 were finally included in this study. The detailed study flowchart is available in Figure 1. The sociodemographic variables are presented in Table 1.

Two hundred and eighteen (71.2%) athletes reported reactions after the first dose of vaccine and 190 (64.6%) after the second. Pfizer/BioNtech was the most used vaccine for the first (83.7%) and the second (90.5%) doses of vaccine (Table 1). About half of the respondents (43.5%) had a duration of vaccine reactions of one to three days for the first dose and 35.4% for the second dose (Table 1). The intensity of vaccine reactions had a median rating of 10 (2.5–40) for the first dose and 10 (0–50) for the second dose (Table 1). The impact of the vaccine reactions on training had a median score of 50 (20–50) for the first dose and 46 (20–50) for the second one (Table 1).

Regarding COVID-19 infection, 37% of the athletes had been diagnosed with COVID-19 (Table 1), and 61% of them reported a negative impact on their performance. The proportion of athletes infected before vaccination reporting a negative impact on their performance was higher (75.0%) than the athletes infected after at least one dose of vaccine (48.2%).

### 3.2. Perceived Change in Physical Performance Following COVID-19 Vaccination

A total of 221 athletes (72.2%) perceived no change in their physical performance after full COVID-19 vaccination (Table 2), while 72 (23.5%) experienced a negative impact, and 13 (4.2%) felt a positive effect after vaccination (Table 2). A negative performance impact was more frequently observed when the duration of the impact following full vaccination lasted between 8 and 30 days (7.5%) or more than 30 days (6.5%) (Table 2).

### 3.3. Factors Associated with Perception of Negative Performance

In the univariate analysis, the athletes who perceived a negative performance impact after three days: tended to be older than 25 (*p* = 0.006) (Table 3), were more likely to practice individual sports (*p* < 0.0001), had a higher level (*p* < 0.0001) and longer duration (*p* < 0.0001) of vaccine reactions after the first and second doses; had a more negative impact on their training (*p* < 0.0001) and had more pressure to get vaccinated (*p* < 0.0001). The sensitivity analysis with an 8-day threshold showed broadly the same results (Appendix A).

After adjusting for all significant variables in the univariate analysis, the logistic regression (Table 4), conducted only in the athletes who received two doses of vaccine (*n* = 294), showed that practicing an individual sport (OR = 5.56 (1.51–27.64)), the duration of the first vaccine reaction longer than three days (OR = 61.58 (5.93–840.91)), the level of second vaccine reactions (OR = 1.03 (1.01–1.06)) and the perceived pressure to get vaccinated (OR = 1.02 (1.01–1.04)) were independently associated with a perceived negative impact on physical performance of more than three days after the full COVID-19 vaccination (Table 4).

### 3.4. Pressure to Get Vaccinated and Perception of Negative Performance

Half of the athletes were at or above a score of 40 (0–80) on the global pressure scale to receive the COVID-19 vaccine (Figure 2). A highly significant difference (*p* < 0.0001) for this global pressure is observed between athletes with a perceived negative impact on performance over more than three days after full vaccination, for whom the global pressure was 80 (60–100) compared to 30 (0–70) for the athletes who did not perceive a negative performance impact or for less than three days. Similarly, the perceived social and professional pressures were significantly higher in the athletes with a perceived negative impact of more than three days (60 (15–80) and 80 (45–100), respectively) than in the athletes with no negative performance impact (10 (0–50) and 50 (0–70), respectively, *p* < 0.0001).

## 4. Discussion

Our study shows that three out of four of the elite athletes from Belgium, France, Luxembourg and Canada perceived that full COVID-19 vaccination had no impact on their physical performance, while 23% of them perceived a negative impact. Specifically, 8% experienced a negative impact of between 8 and 30 days and 7% more than 30 days. Practicing an individual sport and the pressure to get vaccinated contributed to the perception of a performance limitation.

Currently, post-vaccine recommendations for athletes include considering a temporary reduction in training load in the first 48 to 72 h post-vaccine injection, particularly after the second dose [4,14]. This aligns with our findings.

The negative impacts could be explained, at least partially, by the physiological effect of the vaccine itself on the immune system, some acute inflammatory reaction or the possible impact of previous COVID-19 infection [3,4,9]. Placebo or nocebo effects of vaccination can also not be ruled out. Indeed, the subjective and behavioral outcomes of drug use are influenced by the expected effects of the drug [15]. The manipulation of these expected effects has been shown to alter the behavioral and subjective effects [16].

Regarding the type of vaccine, no significant difference was observed between the different vaccines on the perceived performance in our study. A recent study showed that the Pfizer COVID-19 vaccination has minimal effects on the physiological responses to graded exercise in physically active healthy people up to three weeks after the vaccination [17]. However, the authors noted that the slight increases in cardiovascular and neuroendocrine responses to exercise after the vaccine regimen may have implications for elite athletes and are worthy of investigation [17].

It appeared with the logistic regression that the duration (more than three days) of vaccine reactions—and not the level of these reactions—after the first vaccine was significantly associated with the probability to have a negative impact on physical performance for more than three days, while, for the second vaccine, it was the opposite: the level of vaccine reactions was significantly associated but not the duration. This could be partly explained by the fact that more athletes reported the duration of side effects for more than three days after the second vaccine (15% versus 8% after the first vaccine) even though the reported levels of reactions on the VAS scale were similar for both vaccines. The duration of vaccine reactions, more than their intensity, could contribute to the negative performance perception.

We found that perceived pressure to get vaccinated was an independent factor of perceived change in performance: the higher the pressure, the greater the perceived negative impact on performance. This could probably be partly explained by the impact of pressure to be vaccinated on the level and intensity of the reported vaccine reactions. A reporting bias might also contribute to this pattern: the participants who felt more pressure to get vaccinated might have paid more attention to the vaccine reactions. The type of pressure further played a role: professional pressure proved to be stronger than social pressure. This is in line with previously reported findings on Polish elite athletes, for whom coaches had a strong influence on their decision to get vaccinated [10]. It highlights the importance of comprehensive communication between the athlete and their multidisciplinary staff. The impact of social norms (i.e., when most of the entourage is vaccinated, athletes may feel pressure to get vaccinated) should also be acknowledged and considered in future vaccination campaigns.

Our study further suggests that individual sports appear to have a greater impact on the perception of physical performance modification. Individual athletes might be more sensitive to these types of changes, possibly without feedback from teammates to confirm or tone down their perceptions. This is in line with findings from a recent study involving 274 Spanish participants practicing physical activity a minimum of three hours a week, which showed that the participants who practiced sports with others showed a quicker rate of adaptation to somatic symptoms during lockdown [18]. The perception of vaccination or infection-related effects on performance, therefore, appears to differ between individual and team sport athletes.

The effect of COVID-19 itself may have had some impact on performance. In our study, 37% of the athletes had been diagnosed with this disease, and over 61% of them reported an impact on their performance. Importantly, among these athletes, 49% had COVID-19 after at least one dose of vaccine, highlighting the difficulty to assess the independent effect of the disease or the vaccine on performance. However, the higher proportion (75%) of athletes reporting a negative impact on performance following COVID-19 infection before their first vaccination—as opposed to those who report a negative impact on performance following COVID-19 infection after vaccination (48%)—suggests that the infection itself has a more deleterious impact on performance perception than the vaccination. The present study was, however, not designed to answer this specific aspect, and follow-up studies should further investigate it.

The present study encompasses some limitations. The first limitation was the use of a web-based anonymous survey that can induce some reporting and selection bias. Such surveys are, notwithstanding, increasingly praised for their fast, flexible and far-reaching properties for data collection [19]. The 34% response rate further limits the generalizability of our findings to the general elite athletes’ population. This proportion falls between the ones reported in similar studies using web-based surveys with elite-level athletes (14% [10], 66% [20] or not reported [9]). The present survey was international and targeted elite athletes from four countries. However, the participation rate was heterogeneous, and a majority of the participants were from Belgium, affecting the sample characteristics. Second, we were unable to actually measure performance (or use performance indicators), meaning the reported perceptions cannot be crossed with objective performance variables from the field. In addition, we did not assess the duration of the perception of performance modification. We further acknowledge some potential recall bias since the questionnaire was filled-in months after the first dose. However, to our knowledge, these findings are unique and specific to the elite-level population. We used strict inclusion criteria to pin down this specific population among a rather homogenous sample. This also means our results could hardly be extrapolated to the general population.

## 5. Conclusions

Our study shows that 72% of elite athletes perceived that vaccination against COVID-19 had no impact on their performance, while 23% of them experienced a context of perceived reduced performance after full vaccination. Practicing an individual sport and perceived pressure to get vaccinated contributed to performance limitation perceptions. These should be considered in future vaccination campaigns.

## Figures and Tables

**Figure 1 vaccines-11-00796-f001:**
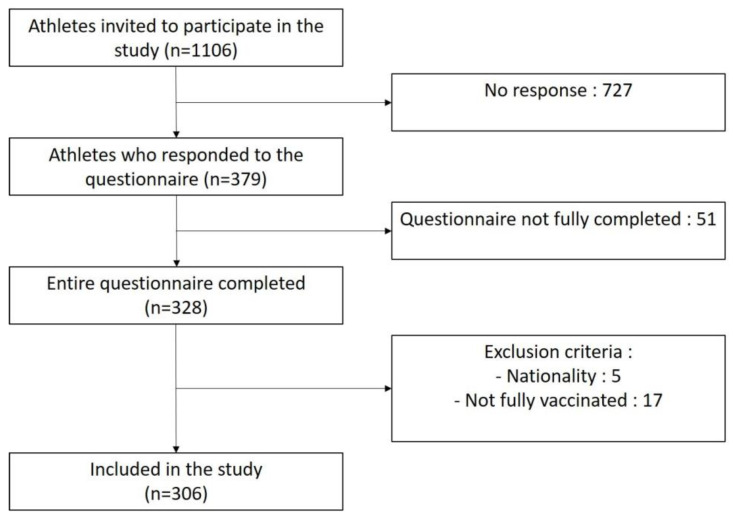
Study flowchart.

**Figure 2 vaccines-11-00796-f002:**
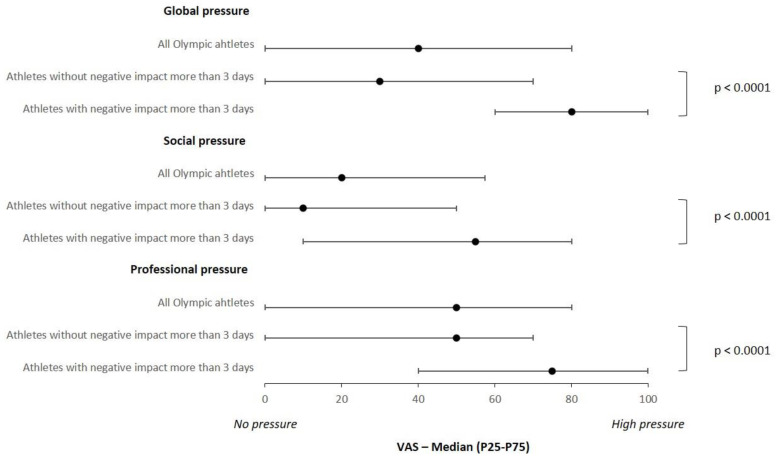
Perceived global, social and professional pressure to get vaccinated on a 100-unit visual analog scale (VAS).

**Table 1 vaccines-11-00796-t001:** Sociodemographic and COVID-19 vaccination characteristics of the included population of elite athletes (*n* = 306).

Characteristics		Number	Frequency (%)	Median (P25–P75)
Sociodemographic				
Age (years)	<18	48	15.7	
	18–25	157	51.3	
	>25	101	33.0	
Nationality	Belgium	178	58.2	
	Canada	49	16.0	
	France	60	19.6	
	Luxembourg	19	6.2	
Sport	Collective	86	28.1	
	Individual	220	71.9	
Sport by week (hours)	Between 5 and 10	16	5.2	
	Between 11 and 20	151	49.4	
	Between 21 and 35	127	41.5	
	Between 36 and 50	12	3.9	
COVID-19 positive	Yes	114	37.3	
	No	192	62.7	
COVID-19 positive before	Yes	58	50.9	
first dose of vaccine	No	56	49.1	
COVID-19 vaccination				
First dose of vaccine (*n* = 306)				
Vaccine type	AstraZeneca	7	2.3	
	Johnson & Johnson	28	9.1	
	Moderna	15	4.9	
	Pfizer/BioNTech	256	83.7	
Level of vaccine reactions (VAS)				10 (2.5–40)
Duration of vaccine reactions	No reaction	88	28.8	
	Less than 1 day	61	19.9	
	Between 1 and 3 days	133	43.5	
	More than 3 days	24	7.8	
Impact of vaccine reactions on training (VAS)				50 (20–50)
Second dose of vaccine (*n* = 294)				
Vaccine type	AstraZeneca	6	2.0	
	Johnson & Johnson	3	1.0	
	Moderna	19	6.5	
	Pfizer/BioNTech	266	90.5	
Level of vaccine reactions (VAS)				10 (0–50)
Duration of vaccine reactions	No side reaction	104	35.4	
	Less than 1 day	42	14.3	
	Between 1 and 3 days	104	35.4	
	More than 3 days	44	14.9	
Impact of vaccine reactions on training (VAS)				46 (20–50)

Note: Six athletes received two doses of the AstraZeneca vaccine, while twelve athletes received the Johnson & Johnson vaccine as a first dose and did not receive a second dose, which explains the difference in the number of respondents between the first and second doses.

**Table 2 vaccines-11-00796-t002:** Perceived impact on physical performance according to the duration of the vaccine reactions (*n* = 306).

	Perceived Impact on Physical Performance		
Duration of Vaccine Reactions			
	No Performance Impact	Negative Impact	Positive Impact
No reaction	221 (72.2%)		
Reaction less than 1 day		2 (0.7%)	1 (0.3%)
Between 1 and 3 days		15 (4.9%)	3 (1.0%)
Between 4 and 7 days		12 (3.9%)	3 (1.0%)
Between 8 and 30 days		23 (7.5%)	1 (0.3%)
More than 30 days		20 (6.5%)	5 (1.7%)

**Table 3 vaccines-11-00796-t003:** Factors associated with the perceived negative impact on physical performance lasting more than 3 days after full COVID-19 vaccination.

Characteristics	No Negative Impact or Negative Impact <3 Days (*n* = 251)	Negative Impact >3 Days (*n* = 55)	*p*-Value
Age (years)			
<18	45 (17.9%)	3 (5.5%)	**0.006**
18–25	132 (52.6%)	25 (45.4%)	
>25	74 (29.5%)	27 (49.1%)	
Sport			
Collective	82 (32.7%)	4 (8.9%)	**<0.0001**
Individual	169 (67.3%)	51 (91.1%)	
Training hours per week			
Between 5 and 10	15 (6.0%)	1 (1.8%)	0.15
Between 11 and 20	129 (51.4%)	22 (40.0%)	
Between 21 and 35	97 (38.6%)	30 (54.6%)	
Between 36 and 50	10 (4.0%)	2 (3.6%)	
Type of vaccine			0.94
AstraZeneca	6 (2.4%)	1 (1.8%)	
Johnson & Johnson	25 (9.9%)	3 (5.5%)	
Moderna	13 (5.2%)	2 (3.6%)	
Pfizer/BioNTech	207 (82.5%)	49 (89.1%)	
First dose of vaccine (*n* = 306)			
Level of vaccine reactions (VAS)	10 (0–40)	30 (10–70)	**<0.0001**
Duration of vaccine reactions			
No vaccine reaction	83 (33.1%)	5 (9.1%)	**<0.0001**
Less than 1 day	57 (22.7%)	4 (7.3%)	
Between 1 and 3 days	106 (42.2%)	27 (49.1%)	
More than 3 days	5 (2.0%)	19 (34.5%)	
Impact on training (VAS)	50 (30–50)	20 (0–50)	**<0.0001**
Second dose of vaccine (*n* = 294)			
Level of vaccine reactions (VAS)	10 (0–30)	70 (40–90)	**<0.0001**
Duration of vaccine reactions			
No vaccine reaction	100 (41.8%)	4 (7.3%)	**<0.0001**
Less than 1 day	38 (15.9%)	4 (7.3%)	
Between 1 and 3 days	88 (36.8%)	16 (29.1%)	
More than 3 days	13 (5.4%)	31 (56.4%)	
Impact on training (VAS)	49 (30–50)	10 (0–30)	**<0.0001**
Injury since vaccination			
Yes	84 (35.1%)	20 (36.4%)	0.86
No	155 (64.9%)	35 (63.6%)	
Pressure to get vaccinated	30 (0–70)	80 (60–100)	**<0.0001**
Positive for COVID-19 before first dose			
Yes	50 (19.9%)	8 (14.5%)	0.36
No	201 (80.1%)	47 (85.5%)	

**Table 4 vaccines-11-00796-t004:** Logistic regression on the probability to have a negative impact on physical performance of more than 3 days after the full COVID-19 vaccination (*n* = 294).

Characteristics	OR (IC95%)	*p*-Value
Age (years) Ref: <18		
18–25	1.54 (0.33–9.34)	0.60
>25	1.89 (0.39–11.73)	0.45
Sport Ref: collective		
Individual	5.56 (1.51–27.64)	**0.02**
First vaccine		
Level of vaccine reactions	0.98 (0.95–1.00)	0.09
Duration of vaccine reaction Ref: No reaction		
Less than 1 day	1.21 (0.20–6.75)	0.83
Between 1 and 3 days	3.47 (0.76–17.43)	0.12
More than 3 days	61.58 (5.93–840.91)	**0.001**
Impact on training	0.99 (0.96–1.01)	0.29
Second vaccine		
Level of vaccine reactions	1.03 (1.01–1.06)	**0.004**
Duration of vaccine reaction Ref: No reaction		
Less than 1 day	0.92 (0.15–5.37)	0.92
Between 1 and 3 days	0.34 (0.06–2.05)	0.23
More than 3 days	0.93 (0.10–8.62)	0.94
Impact on training	0.98 (0.96–1.01)	0.21
Pressure to get vaccinated	1.02 (1.01–1.04)	**0.001**

## Data Availability

The data will be made available upon reasonable request to the corresponding author (geraldine.martens@uliege.be).

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
