# Peer review of "Impact of COVID-19 Vaccination on Short-Term Perceived Change in Physical Performance among Elite Athletes: An International Survey"

_vaccines, 2023, doi:10.3390/vaccines11040796_

Round 1

Reviewer 1 Report

This paper reported the results of an online survey to understand the impact of the SARS-CoV-2 vaccination on elite athletes in Europe countries. The results will be truly helpful for future vaccination for elite athletes; however, I strongly recommend that the article be carefully reorganized with appropriate sub-headers in the Methods and Results sections. In addition,

1) Methods section should be reorganized with sub-headers; for example, participant recruitment, questionnaire development, ethical consideration, outcomes, and statistical analysis.

2) Results section should be reorganized with sub-headers, as well; for example, participant characteristics, change in their physical performance after full COVID-19 vaccination, participant characteristics associated with the perception of negative performance over three days, and the association between the negative-performance perception and pressure elements.

3) The study flow chart is an important element of the article. Figure S1 should appear in the main text rather than supplementary material.

4) Tables S1 and S3 are also should be in the main text, not in supplementary materials since the tables show the analysis of the primary outcome of this study.

5) Figure 1 needs to be modified to show the combination of variables with statistical significance.

5) Lastly, a discussion on the result of the multivariate logistic analysis is needed; there is a difference in the effect of the adverse event between the first and second vaccination. The duration of adverse events is associated with a negative impact on physical performance in the first vaccination; however, the levels of adverse events are in the second. This might be addressed by the descriptive statistics on the levels and durations of adverse events after the first and second vaccination.

Author Response

This paper reported the results of an online survey to understand the impact of the SARS-CoV-2 vaccination on elite athletes in Europe countries. The results will be truly helpful for future vaccination for elite athletes; however, I strongly recommend that the article be carefully reorganized with appropriate sub-headers in the Methods and Results sections. In addition,

 We thank the reviewer for the encouraging feedback.

1) Methods section should be reorganized with sub-headers; for example, participant recruitment, questionnaire development, ethical consideration, outcomes, and statistical analysis.

We reorganized the Methods section with the following sub-headers: study procedures, participant recruitment, questionnaire and analyses.

2) Results section should be reorganized with sub-headers, as well; for example, participant characteristics, change in their physical performance after full COVID-19 vaccination, participant characteristics associated with the perception of negative performance over three days, and the association between the negative-performance perception and pressure elements.

We reorganized the Results section with the following sub-headers: participant characteristics, perceived change in physical performance following COVID-19 vaccination, factors associated with perception of negative performance and pressure to get vaccinated and perception of negative performance.

3) The study flow chart is an important element of the article. Figure S1 should appear in the main text rather than supplementary material.

We followed the reviewer’s suggestion and placed the study flowchart as a figure in the main manuscript.

4) Tables S1 and S3 are also should be in the main text, not in supplementary materials since the tables show the analysis of the primary outcome of this study.

We followed the reviewer’s suggestion and placed Tables S1 and S3 in the main manuscript.

5) Figure 1 needs to be modified to show the combination of variables with statistical significance.

 We suggest an updated version of the figure in the manuscript showing the combination of variables with statistical significance.

6)  Lastly, a discussion on the result of the multivariate logistic analysis is needed; there is a difference in the effect of the adverse event between the first and second vaccination. The duration of adverse events is associated with a negative impact on physical performance in the first vaccination; however, the levels of adverse events are in the second. This might be addressed by the descriptive statistics on the levels and durations of adverse events after the first and second vaccination.

We thank the reviewer for raising this interesting aspect. We added the following excerpt in the discussion section:

“It appeared with the logistic regression that the duration (above three days) of vaccine reactions - and not the level of these reactions - after the first vaccine was significantly associated with the probability to have a negative impact on physical performance for more than three days while for the second vaccine it was the opposite: the level of vaccine reactions was significantly associated and not the duration. This could be partly explained by the fact that more athletes reported a duration of side effects for more than three days after the second vaccine (15% versus 8% after first vaccine) even though the reported levels of reactions on the VAS scale were similar for both vaccines. The duration of vaccine reactions, more than their intensity, could contribute to the negative performance perception.”

Reviewer 2 Report

This article is an interesting cross-sectional study that focuses on the impact of COVID-19 vaccination on short-term perceptions of physical performance among elite athletes and the factors that influence this perception, especially the pressure to vaccinate.

The study may be of interest to the scientific community and is methodologically sound, although there are some aspects that should be reviewed/clarified before publication.

- Why was the gender variable not included among the socio-demographic variables studied? This is a very significant absence. This data should be included in the analysis.

- There is some information in the results that should be clarified.

- Figure S1: Study flowchart, and Tables S1 and S3 should appear in the text of the main article, not in the supplementary material.

I have added the rest of the contributions in the form of comments in the attached document.

Best regards

Author Response

This article is an interesting cross-sectional study that focuses on the impact of COVID-19 vaccination on short-term perceptions of physical performance among elite athletes and the factors that influence this perception, especially the pressure to vaccinate.

 We thank the reviewer for the encouraging feedback.

The study may be of interest to the scientific community and is methodologically sound, although there are some aspects that should be reviewed/clarified before publication.

- Why was the gender variable not included among the socio-demographic variables studied? This is a very significant absence. This data should be included in the analysis.

We agree it is an important factor. To guarantee the anonymous character of the data collection (as elite athletes represent a sensitive population with regards to anonymity), we collected as few sociodemographic data as possible and as requested by the national olympic committees to avoid identification of the participants by crossing their answers.

We added a sentence in the revision:

No additional sociodemographic data was collected as this aspect was restricted to guarantee the anonymous character.

- There is some information in the results that should be clarified.

We clarified this information following the comments provided in the attached document.

- Figure S1: Study flowchart, and Tables S1 and S3 should appear in the text of the main article, not in the supplementary material.

We placed the study flowchart as well as Tables S1 and S3 in the main article.

I have added the rest of the contributions in the form of comments in the attached document.

We incorporated the suggestions in the comments and answered these comments in the attached document.

Round 2

Reviewer 1 Report

The revisions were appropriately made to address the issues pointed out. The grammatical corrections should be made before publication. I recommend adding the name and version of the R packages implemented in this study.

Reviewer 2 Report

The authors have done a very thorough review, incorporating the contributions made in an appropriate manner.
The only thing I do not find very convincing is the explanation that they did not collect the sex of the athletes to ensure anonymity. In any case, I understand that if this variable was not collected at the time, they cannot now incorporate it into the analysis.
For the rest, everything is correct.
Congratulations for the work done.
Best regards